# Maize Lodging Resistance with Plastic Film Removal, Increased Planting Density, and Cultivars with Different Maturity Periods

**DOI:** 10.3390/plants11202723

**Published:** 2022-10-15

**Authors:** Xiyun Zhang, Jun Xue, Ming Tian, Guoqiang Zhang, Bo Ming, Keru Wang, Peng Hou, Ruizhi Xie, Qiuxiang Tang, Shaokun Li

**Affiliations:** 1College of Agronomy, Xinjiang Agricultural University, Urumqi 830052, China; 2Key Laboratory of Crop Physiology and Ecology, Ministry of Agriculture and Rural Affairs, Institute of Crop Sciences, Chinese Academy of Agricultural Sciences, Beijing 100081, China

**Keywords:** maize lodging, plastic film mulch, no-film planting, stalk strength, vertical root-pulling force

## Abstract

While plastic film mulching and proper high-density planting are important methods that can improve maize yield, years of accumulated residual film have created soil pollution and degraded soil, and thus has impeded sustainable agriculture development. Here, we compared the stalk and root lodging resistances of three maize cultivars grown at two planting densities both with (FM) and without (NM) plastic film mulch. Our aim was to provide a theoretical basis that may help assure a future of successful no-film planting with increased planting density. The results showed that, compared with FM, the average dry weight per unit length and bending strength of basal internode decreased for all cultivars at both planting densities in the NM treatment. At 9.0 × 10^4^ plants ha^−1^, the stalk breaking force (SFC) of Xinyu77, KWS9384, and KWS2030 in the NM treatment decreased by 4%, 21%, and 22%, respectively. At 12.0 × 10^4^ plants ha^−1^, SFC of Xinyu77 and KWS2030 increased by 14% and 1%, respectively, while KWS9384 decreased by 10%. Additionally, the root diameter, length, volume, width, depth, and the vertical root-pulling force of maize decreased. Although the lodging resistance of maize grown without film mulch was lower than that of maize grown with it, those adverse effects can be mitigated by selecting suitable cultivars and by using proper high-density planting and appropriate cultivation measures.

## 1. Introduction

Plastic film mulching is an important cultivation measure that enables maize to achieve a high yield with great efficiency. It increases rainfall catch, soil water storage, and water reflux, as well as inhibits soil water evaporation, thus improving the crop’s water use efficiency [1,2,3,4]. At the same time, plastic film mulching can increase soil temperature, shorten the maize growth period, increase plant height and leaf area, and significantly increase biological and economic yields, such as the effective spike number and 1000-grain weight [5,6,7,8]. In addition, plastic film can effectively control weeds and reduce herbicide use [9]. Therefore, for the past 40 years, plastic film mulching has played a positive role in increasing crop yield and quality in China.

However, the residual film accumulated by years of film-mulch planting has contributed to soil pollution that leads to declining soil quality and that hinders the sustainable development of agriculture. In 2017, China’s plastic film use reached 1.437 million tons and covered an area of 18.7 million hectares. The amount of plastic film used is estimated to reach 2 million tons and cover an area of 22 million hectares by 2024 [10]. Unfortunately, common polyethylene mulch film does not degrade under natural conditions [11,12,13]. Because of the increasing input and service life of plastic film, a large amount of it remains in the fields, destroying soil structure, obstructing farming operations, and leading to declining cultivated land quality and reduced crop yields [14,15,16]. It is also responsible for increased greenhouse gas emissions [17,18] and pesticide adsorption, which leads to food safety risks [19].

Approaches aimed at solving the residual mulch film pollution are (1) promoting standardized mulch film production and preventing the use of low quality film on farmland by stopping it at its the source; (2) optimizing the mulching mode by promoting the timely uncovering and scientific use of plastic film; (3) strengthening the research, development, and promotion of residual film recycling machinery and improving the mechanical recycling of plastic film; and (4) increasing the research into degradable mulch film products and developing completely degradable mulch film that will not cause secondary pollution [20].

Besides using mulch film, proper high-density planting is also an important way to improve maize yield, and close planting is currently the simplest and most effective high-yield cultivation technique. Maize yield is dictated by the coordination of individual genetic characteristics, group effects, and environment. With increasing density, the grain yield per unit area increases well until it reaches a certain level, then it decreases as density increases, and there was a quadratic correlation [21,22]. Yang et al. [23] hypothesized that increased maize planting density would reduce the leaf area per plant, the number of grains per ear, and the 1000-grain weight. However, high-density planting significantly increased the maize population number, and because that effectively made up for all the indexes reduced by low-density planting, high-density planting had a higher yield. Therefore, breeding and planting maize cultivars with good density tolerance has become a substantial means for increasing maize yield [24,25]. The density tolerance and yield of maize cultivars are related to their growth periods (i.e., length of time to maturity). Under the same planting density, middle-maturing cultivars have a better yield advantage than early-maturing cultivars do [26], while early-maturing cultivars have higher optimum planting density, better lodging resistance at high planting density, and lower grain moisture content at harvest. Additionally, proper high-density planting can make up for any yield loss caused by the short growth period, and thus high yields are obtained [27,28,29,30]. At the same time, mechanized grain harvesting is becoming more prevalent, and it works best with early-maturing cultivars because their high filling rate and good density tolerance is key to realizing high-yield and high-efficiency mechanized harvesting [31,32].

So, will adopting increased density planting of early maturing cultivars enable no-film cultivation? Will it make up for the yield loss caused by low yield per plant? This model has been applied to cotton [33], but maize has a lodging problem from increased density planting. Additionally, its growth and development process without plastic mulch is affected. Therefore, maize lodging resistance during close planting without film should be investigated.

Maize lodging consists of root lodging and stalk lodging. In root lodging, the plant aboveground inclines or leaves the soil together with the root, possibly because the root of the plant is underdeveloped [34]. Maize root growth and development are closely related to lodging [35], and the quantity and distribution of roots in the soil are related to the ability to resist lodging [36,37]. Root morphology and the dry matter weight of maize are important factors that affect plant anchoring and root lodging resistance [38]. Stalk lodging usually refers to the bending, deformation, or fracture of the basal internodes before the roots lodge, and it is most likely to occur at the late jointing stage and after tasseling [39,40]. Plant, ear, and gravity heights, and basal internode length, diameter, and mechanical strength can effectively evaluate the lodging resistance of maize stalks [41,42]. As planting density increases, plant competition for fertilizer, water, and light intensifies. That hinders growth, makes the stalk grow slenderer, reduces stalk strength, affects the distribution of roots in different soil layers, and increases the lodging risk [43,44,45,46].

We conducted a two-year study in Xinjiang, the largest plastic-film-covered area in China [47]. For each growing season, we set two sowing dates to represent different growing environments. Maize cultivars with different maturities were planted under conventional and increased planting densities and with and without plastic film mulch. The growth period, plant morphology, mechanical stalk strength, and vertical root-pulling force of the plants were compared among treatments. As a result, we determined the differences in the stalk and root lodging resistances of plants grown with and without plastic film mulch and clarified the key factor that affects stalk lodging resistance of close-planted maize grown without film. These results provide a theoretical basis and technical support for successful close planting without film.

## 2. Results

### 2.1. Emergence Rate and Growth Period

There was no significant difference in plant emergence rate between film planting (FM) and no-film planting (NM, Figure 1). Compared with FM, NM had delayed and prolonged growth processes (Figure 2). Specifically, for sowing date 1 (April 18), the NM emergence stage was prolonged by 1–2 d, the silking period by 4–6 d, and maturation by 2–5 d; NM decreased the average daily temperature of the 5 cm soil layer in those periods by 0.7 °C, 0.8 °C, and 1.0 °C, respectively; on the 10 cm soil layer, temperature decreased by 0.6 °C, 1.1 °C, and 1.1 °C, respectively. For sowing date 2 (April 29), those periods were prolonged by 2–3 d, 4–7 d, and 2–5 d, respectively.

### 2.2. Stalk Breaking Force

The applied force needed to break a maize stalk in the NM and FM treatments differed among planting densities and cultivars (Figure 3). When the planting density was 9.0 × 10^4^ plants ha^−1^, all three cultivars in the NM treatment had stalk breaking forces less than those in the FM treatment, and, whereas those of cultivar Xinyu77 were not significantly different, those of KWS9384 and KWS2030 were significantly lower in the NM treatment than in the FM treatment. When the planting density was 12.0 × 10^4^ plants ha^−1^, the stalk breaking forces of Xinyu77 and KWS2030 between the FM and NM treatments were not significantly different, but KWS9384′s breaking force in the NM treatment was significantly lower than that in the FM treatment. However, the stalk breaking forces between cultivars Xinyu77 and KWS9384 under the two planting densities were not significantly different, but they were significantly higher than that of KWS2030. Overall, when the planting density increased from 9.0 × 10^4^ plants ha^−1^ to 12.0 × 10^4^ plants ha^−1^, the stalk breaking force decreased by 13%.

### 2.3. Plant and Basal Internode Morphologies

Compared with FM, the NM plant, ear, and center of gravity heights, and basal internode length increased by 2.9 cm, 2.2 cm, 2.5 cm, and 0.3 cm, respectively; the ear coefficient and center of gravity coefficient increased by 2%; and the internode diameter decreased by 0.72 mm (Table 1). Among cultivars, KWS9384 had the lowest plant, ear, and gravity heights, and ear and gravity coefficients and internode length; KWS2030 had the highest ear height, ear and gravity coefficients, and internode length; Xinyu77 had the highest plant and gravity heights, and internode diameter; and KWS2030 had the smallest internode diameter. At the higher planting density, overall plant and gravity heights, internode length, and internode diameter decreased, while the ear height, ear coefficient, and gravity coefficient increased.

### 2.4. Basal Internode Bending Strength

The bending strengths of the stalk basal internodes of different maize cultivars differed between the NM and FM treatments (Figure 4). While Xinyu77 had no significant difference between the FM and NM treatments under both planting densities, that of KWS9384 in the NM treatment was significantly lower than that in the FM treatment. At the lower density, the basal internode bending strength of KWS2030 in the NM treatment was significantly lower than that of the FM treatment, but there was no significant difference between the FM and NM treatments at the higher density.

Overall, the bending strength between the two densities decreased by 21%. At the lower density, the internode bending strength of KWS2030 was significantly lower than that of Xinyu77 and of KWS9384; however, at the higher density, the bending strength of Xinyu77 was significantly higher than that of the other two cultivars, and that of KWS2030 was significantly lower than that of the other two cultivars.

### 2.5. Dry Weight per Unit length of the Basal Internode

Compared with the dry weight per unit lengths (DWUL) of the basal internodes in the FM treatment, those in the NM treatment differed among densities and cultivars (Figure 5). When the planting density was 9.0 × 10^4^ plants ha^−^^1^, the DWULs of all three cultivars in the NM treatment were less than those in the FM treatment, and that of Xinyu77 in the NM treatment was significantly lower than that in the FM treatment. However, there was no significant difference between the FM and NM treatments for KWS9384 and KWS2030. When the planting density was 12.0 × 10^4^ plants ha^−1^, the DWUL of Xinyu77 did not differ significantly between the two treatments, but those of KWS9384 and KWS2030 were significantly lower in the NM treatment than in the FM treatment. Under the two densities, the DWUL did not differ significantly between Xinyu77 and KWS9384, but both were significantly higher than that of KWS2030.

### 2.6. Vertical Root-Pulling Force

When the planting density was 9.0 × 10^4^ plants ha^−^^1^, the vertical root-pulling forces of all three cultivars in the NM treatment were significantly lower than those in the FM treatment (Figure 6). However, when the planting density was increased to 12.0 × 10^4^ plants ha^−1^, the vertical root-pulling forces of Xinyu77 and KWS2030 between the FM and NM treatments were not significantly different. However, as with the lowest density, the vertical root-pulling force of KWS9384 in the NM treatment remained significantly lower than that in the FM treatment. Additionally, the vertical root-pulling force of KWS2030 was significantly lower than that of Xinyu77, but not significantly different than that of KWS9384.

### 2.7. Root Morphology

The root depth and width and total root dry weight decreased by 0.6 cm, 2 cm, and 1.2 g, respectively, in the NM treatment compared with the FM treatment (Table 2), and total root length, surface area, diameter, and volume decreased by 1076.3 cm, 204.9 cm^2^, 0.5 mm, and 4.7 cm^3^, respectively. Xinyu77 had the largest total dry root weight, and root depth, surface area, diameter, and volume; KWS9384 had the largest total root length and the smallest root width; and KWS2030 had the largest root width, but the smallest total dry root weight, total root length, root depth, surface area, diameter, and volume. At the higher planting density, total dry root weight, total root length, root depth, width, surface area, diameter, and volume decreased for all cultivars.

### 2.8. Factors That Influenced the Maize Stalk Breaking Force

Path correlation analysis (Figure 7) showed that the stalk breaking force was positively correlated with the DWUL and the bending strength of the basal internode, and negatively correlated with ear height, gravity height, and basal internode length. However, the DWUL of the basal internode was positively correlated with bending strength and negatively correlated with basal internode length, and ear height was positively correlated with gravity height. Meanwhile the vertical root-pulling force was positively correlated with root diameter, volume, width, and depth, but not with root length.

## 3. Discussion

Maize stalk lodging is related to the plant morphology and mechanical strength of the basal internodes, and, because the stalk breaking force comprehensively reflects the entire plant’s physical integrity, cultivars with higher stalk breaking forces have lower risks of breaking in the field [48,49]. Plants with longer basal internodes have higher ear height and gravity height, and thus a higher lodging risk than plants with shorter and thicker basal internodes [50,51]. The mechanical strength of the stalk basal internode, measured as skin puncture strength, crushing strength, and bending strength, is an important factor that determines stalk breaking susceptibility, and all those measures are negatively correlated with the stalk breaking rate in the field [52,53]. Maize root lodging resistance is closely related to its anchoring ability [54]. The maximum tensile force and tensile strength are important measures of root tensile capacity and of root mechanical indexes [55]. The number of roots, total root volume, angle between roots and vertical direction, root diameter, and vertical root-pulling force are closely related to lodging [56,57,58]. Here, we showed that on the one hand basal internode DWUL affected the stalk breaking force by affecting its mechanical strength, while on the other hand ear height affected the stalk breaking force by affecting gravity height. Additionally, concerning maize root lodging resistance, root diameter and length affected the vertical root-pulling force through root volume, and root distribution in the soil (root width and root depth) also affected the vertical root-pulling force (Figure 7).

Compared with NM, FM increases soil temperature and accelerates maize growth [59]. In this study, we found that the whole growth period and the silking period of maize grown without film mulch were shortened by 2–5 d and 4–7 d, respectively, compared with those grown with film mulch, thus indicating that FM impacted the growth period most before flowering. Along with soil temperature, light period and ambient temperature also affect maize ear height, and Liu et al. [60] showed that maize ear height is positively correlated with light period. Here, we found that, compared with FM, the growth process under NM was slower, so the light duration during internode development was higher, and thus the ear height and gravity height each increased by 3%. Hou et al. [61] showed that the diurnal temperature range is significantly positively correlated with the dry matter accumulation of aboveground parts. In this study, compared with FM, the temperature difference in the basal internode dry matter accumulation process in the NM treatment was higher, and that is not conducive to internode dry matter accumulation. Consequently, the basal internode DWUL and the basal internode bending strength decreased by 10% and by 7%, respectively, in the NM treatment compared to the FM treatment. Additionally, the increased center of gravity height and the decreased basal internode bending strength contributed to reducing the breaking force by 8%. Considering root lodging resistance, we showed that, compared with FM, root diameter and root length in the NM treatment decreased by 9% and 21%, respectively, resulting in a 21% reduction in root volume. At the same time, the NM treatment’s root depth and root width decreased by 3% and 11%, respectively, compared to those of the FM treatment, and those root changes resulted in a vertical root-pulling force decrease of 16%. To summarize, the lodging resistances of both stalks and roots in the NM treatment was lower than those in FM treatment, and thus the lodging risk is greater.

Both cultivar and planting density affect maize lodging resistance [62]. Our results showed that as planting density increased from 9.0 × 10^4^ plants ha^−1^ to 12.0 × 10^4^ plants ha^−1^, the stalk breaking forces of maize cultivars Xinyu77, KWS9384, and KWS2030 decreased by 17%, 12%, and 9%, respectively, and their vertical root-pulling forces decreased by 14%, 13%, and 5%, respectively. Those results are in line with previous findings that have shown that as planting density increases, both stalk and root lodging resistance decrease [40,45,46]. However, the differences in both lodging resistances between the NM and FM treatments differed among planting densities and cultivars. At the lower density (9.0 × 10^4^ plants ha^−1^) and compared with the FM treatment, the stalk breaking force of Xinyu77, KWS9384, and KWS2030 in the NM treatment decreased by 4%, 21%, and 22%, respectively, but under the higher density (12.0 × 10^4^ plants ha^−1^), Xinyu77 and KWS2030 increased by 14% and 1%, respectively, while KWS9384 decreased by 10%. Thus, we found that under close planting without film, the lodging resistance of early-maturing cultivars increased slightly. Therefore, given proper high-density planting of suitable cultivars, no-film planting may not reduce maize lodging resistance. This study was conducted in only two years. It would be appropriate to treat this as a pilot study. In the future, it is necessary to further study other maize areas and a longer growth season to confirm the results obtained. Given that the lodging resistance of maize stalks and roots may be reduced after no-film planting, the following cultivation measures may be used to reduce those impacts. (1) Because the sowing date is closely related to lodging [63], sow at the right time. As the sowing date is delayed, the increased temperature causes accelerated vegetative development that may enable lodging. However, early sowing may inhibit vegetative growth and thus reduce lodging [64]. (2) Increase potassium fertilizer application because potassium improves stalk strength and promotes root growth [65]. Potassium fertilizer should be applied early as a seeding fertilizer during sowing or as a seedling fertilizer after emergence. (3) From the seedling to the jointing stage, watering should be properly controlled and seedlings should be spaced properly so the plants grow healthily [66]. (4) Because timely use of plant growth regulators can increase the lodging resistance of maize plants [67], application of those chemicals in the maize V7-V9 stages can promote root development, strengthen root activity, increase soil fixation, shorten and thicken basal internodes, thicken the stalk wall, and enhance lodging resistance [66]. (5) Adequate pest control is also necessary to control lodging and to increase yield. Seeds may be dressed or pesticides may be applied during the growing period to control maize stalk infectious diseases such as sheath blight, stalk rot, and bacterial wilt, and to control stalk borers such as the maize borer [67,68].

## 4. Materials and Methods

### 4.1. Experimental Design and Management

The experiment was conducted in Qitai Farm, Xinjiang, China in 2021 and 2022 (43°50′ N, 89°46′ E, altitude 1020 m). From 2011 to 2021, the average annual rainfall during the maize-growing season (April–October) was 168.6 mm, the daily average temperature was 17.3 °C, the sunshine hours were 1693.3 h, the accumulated temperature of at least 10 °C over the whole year was 3160–3499.5 °C, and the frost-free duration was 156–181 d. The loam soil had 65.4 mg kg^−1^ alkali-nitrogen, 51.0 mg kg^−1^ available phosphorus, 106.2 mg kg^−1^ available potassium, 15.7 g kg^−1^ organic matter, and a pH of 8.0.

The split-plot experimental design used planting density (9.0 × 10^4^ and 12.0 × 10^4^ plants ha^−1^) as the main plot factor and maize variety and film covering modes as subplot factors. Three representative hybrid maize cultivars with different growth periods were used in this study: KWS2030 (early maturing), KWS9384 (middle maturing), and Xinyu77 (late maturing). The two film covering modes were film-used (placed in the narrow rows) and not used, and the two sowing dates in both years were April 18 (sowing date 1) and April 29 (sowing date 2). There were 24 treatments in total, and each treatment was set up with three replications. Plants were seeded in alternating wide–narrow row patterns (alternating row spaces of 70 and 40 cm, respectively) with drip irrigation tape in the narrow rows. One day after sowing, 15 mm of water was applied to assure uniform, rapid germination. Over each entire growing season, the plots were irrigated 12 times and the total irrigation amount was 5400 m^3^ ha^−1^ to maintain good soil moisture content. Before sowing, basal fertilizers were applied (45 kg ha^−1^ N, 75 kg ha^−1^ P_2_O_5_, and 37.5 kg ha^−1^ K_2_O), and 309 kg ha^−1^ N, 136.5 kg ha^−1^ P_2_O_5,_ and 150 kg ha^−1^ K_2_O were applied through drip irrigation during the growth period. During the V6-V8 maize growth stages, 600 mL ha^−1^ of the plant growth regulator, DA-6 Ethephon (Jinan, China), was applied. Pesticides were applied as needed to control insect populations and weeds were periodically removed by hand.

### 4.2. Sampling and Measurements

#### 4.2.1. Growth Period

The dates of key growth stages (sowing, emergence, nine-leaf, twelve-leaf, flowering, and maturity) were accurately recorded for each treatment.

#### 4.2.2. Plant and Internode Measurements and Strengths

Four days after flowering (VT + 4 d), five maize plants with the same growth were randomly selected from each plot. While still growing in the field, we used a YYD-1 stalk strength tester (Zhejiang Top Instrument Co., Ltd., Hangzhou, China) to determine each plant’s breaking force, which is the minimum force required to break the maize stalk [69]. After that test, the same plants were cut at ground level, and their plant and ear heights were measured. The lengths of the third, fourth, and fifth base elongation internodes were measured with a ruler. The internode diameters were measured using a Vernier caliper at the narrowest (d_1_) and the widest (d_2_) sides of each internode’s midpoint, and each internode diameter was calculated as (d_1_ + d_2_)/2. Then, the bending strength of the fifth internode was measured with the YYD-1 stalk strength meter’s U-shaped probe set perpendicular to the long diameter direction of the middle ellipse. Immediately after measuring the fifth internode’s bending strength, the three basal internodes were excised from each plant and placed into paper bags. They then underwent enzyme deactivation at 105 °C for 30 min followed by drying at 80 °C to a constant, dry weight. The DWUL of each of the third, fourth, and fifth internodes was then calculated by dividing the dry weight of each internode by its length.

#### 4.2.3. Vertical Root-Pulling Force

Five plants at growth stage V12 were randomly selected in each plot and, while they were still in the ground, the VRPF of each plant was measured by a tensile meter (Beijing North Shouheng Electronic Technology Co., LTD., Beijing, China). The roots of the same uprooted plants were rinsed with water and then the depth and width of each plant’s main roots were measured. The roots were then scanned at all levels with a V800 scanner (Epson, Jakarta, Indonesia) and the analysis program WinRhizoProVision 5.0 (Regent Instruments, Québec City, QC, Canada) was used to obtain root length, diameter, surface area, and volume for each plant. Finally, the roots were enzyme-deactivated at 105 °C for 30 min and then dried at 80 °C to constant weight and the dry weights were recorded.

### 4.3. Statistical Analysis

Statistical analyses were performed using Predictive Analytics Software (PASW) version 23.0 (IBM SPSS, Somers, New York, NY, USA). Data from each sampling date were analyzed separately. Means for datasets with three or more groups were tested using least significant difference tests at the α = 0.05 level, where alpha is the significance level of the test. Pearson correlations helped identify interrelationships among measured parameters, and path correlation analyses of stalk breaking force, VRPF, plant morphology traits, internode mechanical strength, dry matter accumulation, and root morphology were conducted to better understand causal relationships.

## 5. Conclusions

Compared with FM, NM resulted in higher maize ear and gravity heights, decreased basal internode DWUL and mechanical strength, and reduced stalk breaking force. NM also reduced root diameter and length, thus causing reduced root volume. Additionally, the distribution range of the roots in the soil and the root anchoring ability were reduced. Under the higher density planting and in the NM treatment, the range of variation of both stalk and root lodging resistance differed according to cultivar. Therefore, if suitable maize cultivars are combined with suitable cultivation and control measures, both stalk and root lodging resistance may be avoided in no-film planting.

## Figures and Tables

**Figure 1 plants-11-02723-f001:**
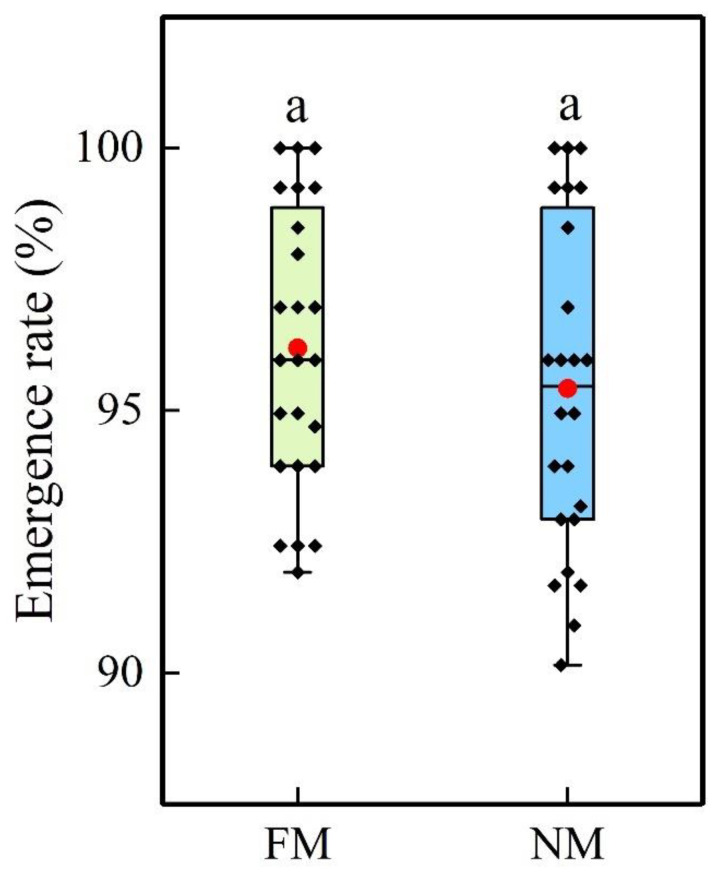
Comparison of emergence rates of film (FM) and no-film (NM) planting. The boxes represent the interquartile ranges (IQR), the whiskers indicate reasonable sample borders (Tukey tests), the horizontal line in each box indicates the median, and the red dot indicates the mean. Different lowercase letters indicate significant differences (*p* < 0.05, *n* = 24).

**Figure 2 plants-11-02723-f002:**
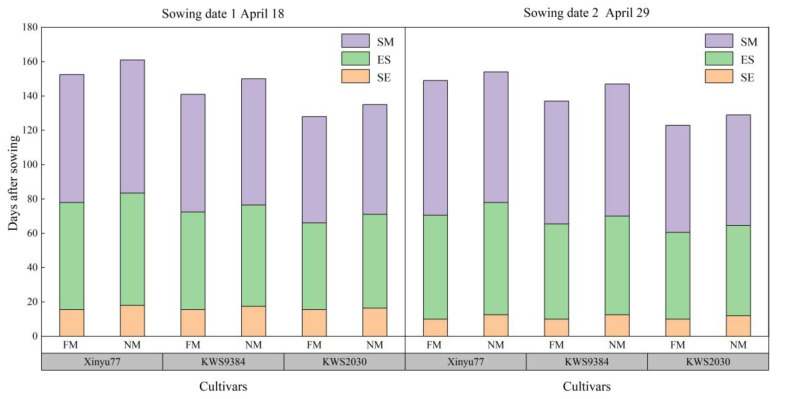
Maize growth periods using film mulch (FM) and no film (NM) planting with two sowing dates in 2021 and in 2022. SE, from sowing to emergence, ES, from emergence to silking, SM, from silking to physiological maturity. Data are means, *n* = 5.

**Figure 3 plants-11-02723-f003:**
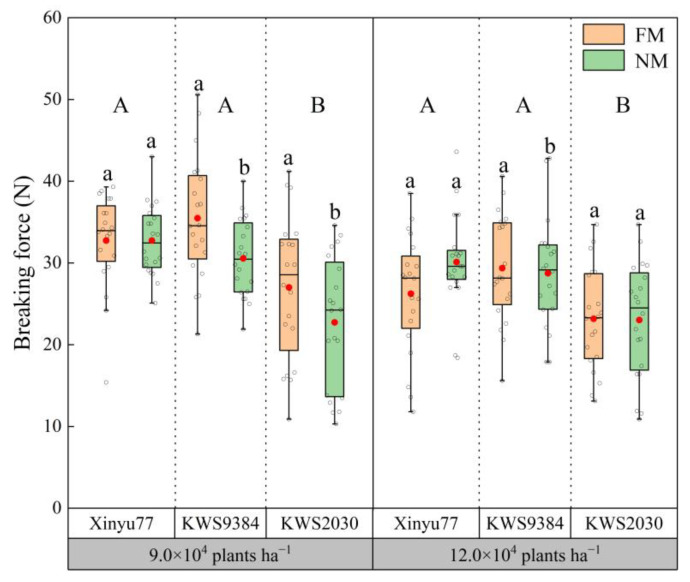
Differences in the stalk breaking forces between film mulch (FM) and no-film (NM) planting under different planting densities and cultivars. The boxes represent IQRs, the whiskers indicate reasonable sample borders (Tukey tests), the horizontal line in each box indicates the median, and the solid red dot shows the mean. Different lowercase letters indicate significant differences (*p* < 0.05) between the FM and NM treatments. Different capital letters indicate significant differences (*p* < 0.05) among cultivars.

**Figure 4 plants-11-02723-f004:**
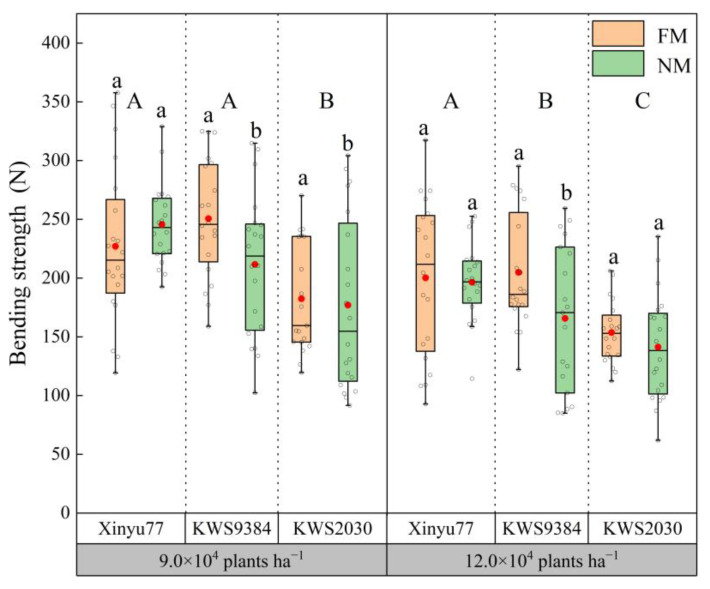
Differences in the basal internode bending strengths between film mulch (FM) and no-film (NM) planting under different planting densities and cultivars. The boxes represent IQRs, the whiskers indicate reasonable sample borders (Tukey tests), the horizontal line in each box indicates the median, and the solid red dot shows the mean. Different lowercase letters indicate significant differences (*p* < 0.05) between the FM and NM treatments. Different capital letters indicate significant differences (*p* < 0.05) among cultivars.

**Figure 5 plants-11-02723-f005:**
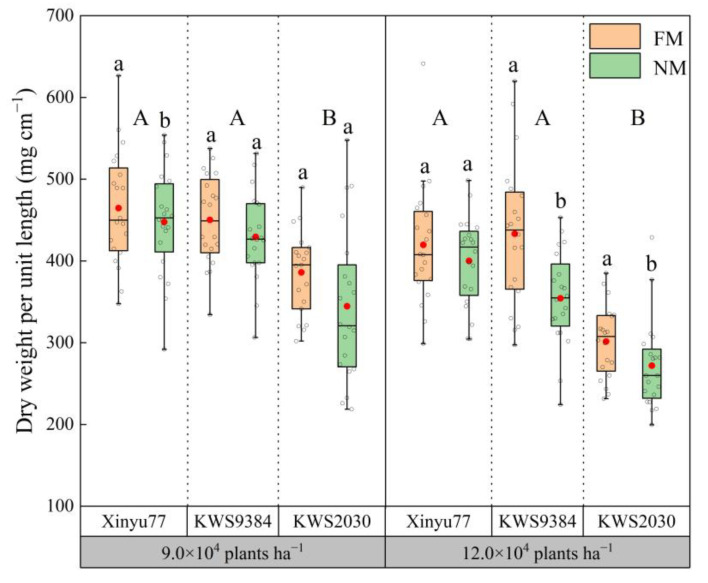
Differences of dry weight per unit lengths between film mulch (FM) and no-film (NM) planting under different plant densities and cultivars. The boxes represent IQRs, the whiskers indicate reasonable sample borders (Tukey tests) the horizontal line in each box indicates the median, and the solid red dot shows the mean. Different lowercase letters indicate significant differences (*p* < 0.05) between the FM and NM treatments. Different capital letters indicate significant differences (*p* < 0.05) among cultivars.

**Figure 6 plants-11-02723-f006:**
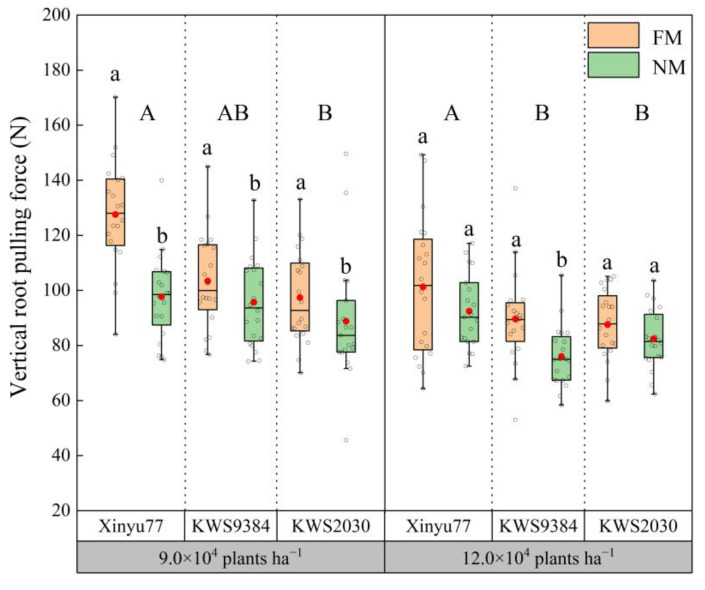
Differences among the vertical root-pulling forces between film mulch (FM) and no-film (NM) planting under different planting densities and cultivars. The boxes represent IQRs, the whiskers indicate the reasonable sample borders (Tukey tests), the horizontal line in each box indicates the median, and the solid red dot shows the mean. Different lowercase letters indicate significant differences (*p* < 0.05) between the FM and NM treatments. Different capital letters indicate significant differences (*p* < 0.05) among cultivars.

**Figure 7 plants-11-02723-f007:**
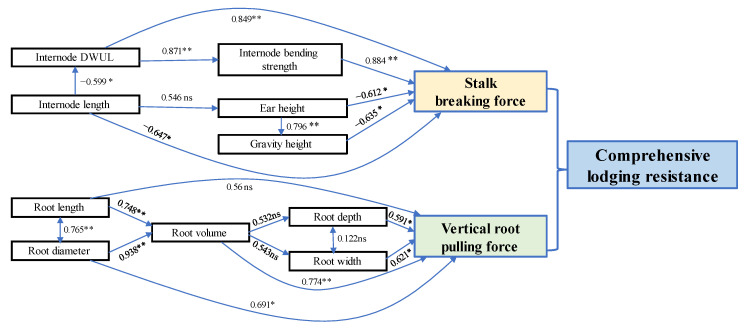
Path correlation analysis of the stalk breaking force, plant morphology traits, stalk mechanical strength, and morphological root traits (*n* = 12). An arrow that points from one variable to another means that the first variable affects the second, and a double-headed arrow means that the two variables covary. Numbers on the lines are path coefficients. * *p* < 0.05; ** *p* < 0.01; ns, no significance.

**Table 1 plants-11-02723-t001:** Differences of maize plant morphology and basal internode morphology between film mulch (FM) and no-film (NM) planting.

Planting Density (Plants ha^−1^)	Cultivar	Treatments	Plant Height (cm)	Ear Height (cm)	Coefficient of Ear Height (%)	Gravity Height (cm)	Coefficient of Gravity Height (%)	Internode Length (cm)	Internode Diameter (mm)
9 × 10^4^	Xinyu77	FM	263.5 (0.2) ^b^	79.0 (1.4) ^b^	31.0 (0.7) ^ab^	86.4 (0.7) ^b^	33.1 (0.4) ^a^	45.7 (0.5) ^c^	21.3 (0.3) ^a^
		NM	268.9 (0.6) ^a^	83.2 (0.8) ^a^	31.4 (0.5) ^ab^	90.0 (0.2) ^a^	33.5 (0) ^a^	45.3 (0.3) ^c^	20.7 (0.2) ^ab^
	KWS9384	FM	259.3 (0.7) ^c^	75.6 (0.3) ^c^	30.1 (0.5) ^b^	81.0 (0.6) ^d^	31.6 (0.3) ^b^	37.8 (0.8) ^e^	20.2 (0.3) ^bc^
		NM	257.3 (0.9) ^c^	76.6 (0.5) ^c^	30.6 (0.5) ^ab^	85.6 (0.9) ^bc^	33.0 (0.5) ^a^	41.9 (0.3) ^d^	19.2 (0.4) ^d^
	KWS2030	FM	257.5 (1.3) ^c^	79.7 (0.9) ^b^	31.3 (0.4) ^ab^	83.8 (0.6) ^c^	32.5 (0.2) ^a^	57.3 (0.5) ^a^	19.5 (0.2) ^cd^
		NM	258.0 (0.3) ^c^	82.4 (0.4) ^a^	32.2 (0.3) ^a^	84.7 (0.4) ^bc^	33.1 (0.2) ^a^	51.8 (0.7) ^b^	18.2 (0.3) ^e^
12 × 10^4^	Xinyu77	FM	254.2 (0.9) ^b^	81.5 (1.7) ^bc^	31.9 (0.7) ^b^	85.1 (0.8) ^b^	33.2 (0.2) ^b^	44.2 (0.5) ^c^	20.1 (0.2) ^a^
		NM	263.4 (1.4) ^a^	84.9 (0.6) ^ab^	32.9 (0.7) ^ab^	87.4 (0.5) ^a^	32.9 (0.5) ^b^	45.5 (0.8) ^c^	19.8 (0.2) ^ab^
	KWS9384	FM	249.8 (0.4) ^c^	81.7 (0.8) ^bc^	32.4 (0.6) ^b^	82.5 (0.4) ^c^	32.9 (0.1) ^b^	39.3 (0.3) ^d^	19.2 (0.2) ^b^
		NM	247.6 (1.1) ^c^	78.9 (1.6) ^c^	32.2 (0.8) ^b^	82.2 (0.6) ^c^	32.7 (0.4) ^b^	39.5 (0.5) ^d^	18.3 (0.3) ^c^
	KWS2030	FM	248.6 (0.9) ^c^	83.5 (0.3) ^b^	33.4 (0.1) ^ab^	83.2 (0.4) ^c^	33.4 (0.1) ^b^	52.4 (0.6) ^b^	17.0 (0.2) ^d^
		NM	255.0 (0.9) ^b^	88.1 (0.4) ^a^	34.6 (0.1) ^a^	87.3 (0.3) ^a^	34.4 (0.2) ^a^	54.5 (0.9) ^a^	16.8 (0.3) ^d^
Planting density (plants ha^−1^)		9 × 10^4^	260.8	79.4	31.1	85.3	32.8	46.6	19.9
		12 × 10^4^	253.1	83.1	32.9	84.6	33.3	45.9	18.5
Mulch treatment		FM	255.5	80.2	31.7	83.7	32.8	46.1	19.55
		NM	258.4	82.4	32.3	86.2	33.3	46.4	18.83
Cultivar		Xinyu77	262.5	82.2	31.8	87.2	33.2	45.2	20.48
		KWS9384	253.5	78.2	31.3	82.8	32.6	39.6	19.23
		KWS2030	254.8	83.4	32.9	84.8	33.4	54.0	17.88

Different lowercase letters indicate a significant difference (*p* < 0.05) between different film covering treatments and cultivars at the same planting density. Data are means with standard error.

**Table 2 plants-11-02723-t002:** Differences of maize root morphology between film mulch (FM) and no-film (NM) planting.

Plant Density (Plants ha^−1^)	Cultivar	Treatments	Root Depth (cm)	Root Width (cm)	Total Dry Root Weight (g)	Total Root Length (cm)	Root Surface Area (cm^2^)	Root Diameter (mm)	Root Volume (cm^3^)
9 × 10^4^	Xinyu77	FM	24.4 (0.2) ^a^	21.7 (0.4) ^a^	9.0 (0.4) ^a^	6772.8 (774.3) ^b^	1548.9 (190.0) ^a^	9.0 (0.5) ^a^	40.9 (2.6) ^a^
		NM	22.6 (1.0) ^b^	16.8 (0.2) ^d^	6.8 (0.4) ^bc^	2491.3 (224.6) ^d^	642.2 (32.7) ^c^	5.0 (0.5) ^c^	13.6 (0.3) ^c^
	KWS9384	FM	22.3 (0.6) ^b^	17.8 (0.3) ^cd^	7.4 (0.2) ^b^	5710.0 (269.1) ^b^	1131.5 (48.8) ^b^	4.7 (0.3) ^c^	17.7 (0.3) ^c^
		NM	20.3 (0.3) ^c^	16.9 (0.5) ^d^	6.8 (0.3) ^bc^	8616.4 (637.0) ^a^	1693.4 (52.3) ^a^	7.3 (0.5) ^b^	27.1 (1.0) ^b^
	KWS2030	FM	19.4 (0.4) ^c^	20.2 (0.6) ^b^	6.1 (0.3) ^cd^	3934.4 (388.5) ^c^	766.5 (29.4) ^c^	4.1 (0.4) ^c^	16.7 (0.7) ^c^
		NM	20.4 (0.4) ^c^	18.7 (0.5) ^c^	5.7 (0.4) ^d^	3675.3 (199.8) ^cd^	867.4 (38.2) ^c^	5.4 (0.4) ^c^	16.7 (1.4) ^c^
12 × 10^4^	Xinyu77	FM	21.2 (0.4) ^ab^	18.3 (0.5) ^ab^	7.4 (0.3) ^a^	6773.8 (780.4) ^a^	1561.0 (143.9) ^a^	6.7 (0.4) ^a^	28.8 (0.4) ^a^
		NM	20.0 (0.4) ^b^	16.3 (0.1) ^cd^	4.8 (0.3) ^b^	3356.8 (168.3) ^bc^	841.8 (51.7) ^b^	4.5 (0.4) ^b^	21.9 (1.8) ^b^
	KWS9384	FM	20.6 (0.9) ^ab^	16.6 (0.3) ^c^	4.8 (0.4) ^b^	4593.2 (679.5) ^b^	955.8 (72.4) ^b^	3.8 (0.5) ^b^	14.5 (0.7) ^c^
		NM	22.1 (0.5) ^a^	15.3 (0.6) ^d^	3.9 (0.1) ^c^	3689.5 (199.1) ^bc^	759.0 (36.1) ^b^	3.7 (0.3) ^b^	13.4 (0.1) ^c^
	KWS2030	FM	17.2 (0.3) ^c^	19.0 (0.4) ^a^	4.9 (0.3) ^b^	3581.8 (150.8) ^bc^	782.1 (38.4) ^b^	4.0 (0.3) ^b^	14.1 (0.3) ^c^
		NM	16.1 (0.6) ^c^	17.3 (0.1) ^bc^	4.5 (0.2) ^bc^	3078.9 (410.3) ^c^	712.5 (64.3) ^b^	3.5 (0.3) ^b^	11.8 (1) ^c^
Plant density (plants ha^−1^)		9 × 10^4^	21.6	18.7	7.0	5200.0	1108.3	5.9	22.1
		12 × 10^4^	19.5	17.1	5.1	4179.0	935.4	4.4	17.4
Mulch treatment		FM	20.9	18.9	6.6	5227.7	1124.3	5.4	22.1
		NM	20.3	16.9	5.4	4151.4	919.4	4.9	17.4
Cultivar		Xinyu77	22.1	18.3	7.0	4848.7	1148.5	6.3	26.3
		KWS9384	21.3	16.7	5.7	5652.3	1134.9	4.9	18.2
		KWS2030	18.3	18.8	5.3	3567.6	782.1	4.3	14.8

Different lowercase letters indicate a significant difference (*p* < 0.05) between different film covering treatments and cultivars at the same planting density. Data are means with standard error.

## Data Availability

Not applicable.

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
