# Peer review of "Maize Lodging Resistance with Plastic Film Removal, Increased Planting Density, and Cultivars with Different Maturity Periods"

_plants, 2022, doi:10.3390/plants11202723_

Round 1

Reviewer 1 Report

The research presented in the manuscript provides information on the resistance of corn to lodging after removing the film and increasing the planting density for three varieties with different maturity.

The main achievement is to indicate the possibility of reducing adverse natural phenomena by selecting the right varieties, applying the right planting density and performing the right tillage treatments.

The authors have conducted meticulous research, which is clearly described and illustrated in the manuscript. In my opinion, the paper is interesting, makes an important contribution to the literature on the subject and is in line with the profile of the journal Plants. It can be published with suggested minor corrections:

1. Table 1

Why are the lowercase letters next to the parentheses if the difference between the averages? Usually significant differences are presented as superscripts - I encourage you to keep this description technique

 Line 163 - "Different lowercase letters indicate a significant difference (p = 0.05)" here in parentheses should be alpha=0.05 or p < 0.0, I think..

Line 165 - "Data are means (SD)." The abbreviation SD is not explained

 2. line 386: I propose instead of "at the p < 0.05 level to give "at the α = 0.05", where alpha is the significance level of the test.

 3. Do I understand correctly that the experimental set-up used assumed 5 replications?

 4. The study was conducted in only 2 years. It would be appropriate to treat them as a pilot study. Therefore, are longer studies planned to confirm the results obtained?

Best Regards

Author Response

Dear reviewer,

Thanks for the comments on our manuscript  “Maize Lodging Resistance with Plastic Film Removal, Increased Planting Density, and Cultivars with Different Maturity Periods” (No: plants-1956231).

We appreciate and accept the modification suggestions and have revised the manuscript accordingly. The revised parts are shown in the original manuscript using red text. The detailed responses to your comments are presented as follows:

  1. Table 1:Why are the lowercase letters next to the parentheses if the difference between the averages? Usually significant differences are presented as superscripts - I encourage you to keep this description technique

Line 163 - "Different lowercase letters indicate a significant difference (p = 0.05)" here in parentheses should be alpha=0.05 or p < 0.0, I think..

Line 165 - "Data are means (SD)." The abbreviation SD is not explained

Reponses: Yes. We have revised it according to your suggestion. The detailed correction is shown in Table 1, Table 2, Line 164-166, and Line 234-236.

  1. line 386: I propose instead of "at the p < 0.05 level to give "at the α = 0.05", where alpha is the significance level of the test.

Reponses: Yes. According to your suggestion, we have added the more information and references about bending strength tests. The detailed correction is shown in Line 393.

  1. Do I understand correctly that the experimental set-up used assumed 5 replications?

Reponses: Yes. In our field experiment, each treatment was set up with three replications. The detailed correction is shown in Line 346-347.

  1. The study was conducted in only 2 years. It would be appropriate to treat them as a pilot study. Therefore, are longer studies planned to confirm the results obtained?

Reponses: Thank you for your proposal, and we will seriously consider the suggestion that further study in other maize areas and more longer growth season to confirm the results obtained.The detailed correction is shown in Line 310-313.

Reviewer 2 Report

This manuscript was tastefully constructed.  The authors did a great job of detailing a problem in corn production.....scraps of years worth of plastic film that seem to be accumulating in the soil.  I assume there was some reason why this row cover was used originally? Heat the soil, or more likely keep weeds from coming up to compete with the corn. Well written manuscript too, excellent language skills. 

Author Response

Dear reviewer,

Thanks for the comments on our manuscript “Maize Lodging Resistance with Plastic Film Removal, Increased Planting Density, and Cultivars with Different Maturity Periods” (No: plants-1956231).

We appreciate and accept the modification suggestions and have revised the manuscript accordingly. The revised parts are shown in the original manuscript using red text. The detailed responses to your comments are presented as follows:

  1. Line 32: “has the ability to increase” should be changed to “increases”.

Reponses: Yes. We have revised it according to your suggestion. The detailed correction is shown in Line 32.

  1. Line 73: “early-maturing” should be changed to shorter?

Reponses: Thank you very much. Early-maturing this statement is correct.

  1. Line 62: “line correlation”, it isn’t linear, part of it is then it levels off.

Reponses: Yes. We have revised it according to your suggestion. The detailed correction is shown in Line 62-64.

  1. Line 112-117: Rain amount of time can make a difference. It depends a lot of growing season. H2O available between VT-R1.

Reponses: Thank you very much. The study area is semi-arid area. From 2011 to 2021, the average annual rainfall during the maize growing season (April–October) was 168.6 mm. In this study, over each entire growing season, the plots were irrigated 12 times and the total irrigation amount was 5400 m3 ha-1 to maintain good soil moisture content. These details were showed in Line 334-335 and Line 350-351.

  1. Figure2: Film mulch increase soil temp. very interesting temp.

Reponses: Yes. NM decreased soil temperature. We added the average daily temperature of 5 cm and 10 cm soil layer in the results section. The detailed correction is shown in Line 115-117.

  1. Figure3: 0×104 plants ha-1-KWS9384?

Reponses: Yes. I re-analyzed the data of KWS9384. 12.0×104 plants ha-1, and the results showed that FM and NM were significantly different.

  1. Line 252 and Line 311: since substitute for because.

Reponses: Yes. We have revised it according to your suggestion. The detailed correction is shown in Line 254 and Line 315.

  1. Line 314-315: “Increase potassium fertilizer application because potassium improves stalk strength” under developed roots in cold soil.

Reponses: Yes. According to your suggestion, we have added the more information and references on the effects of potassium fertilizer on roots. The detailed correction is shown in Line 319.

  1. Line 348: “DA-6 Ethephon (China Agrotech, Shanxi, China)” is the standard?

Reponses: Yes. Using plant growth regulator is the standard when plant density higher than 9.0 ×104 plants ha-1.

Reviewer 3 Report

This paper presents research on demonstrating differences in stalk and root lodging resistance of crops grown with and without film. The authors have explained the important factors affecting stalk lodging resistance of crop plants.

In my opinion, the reviewed article presents a novel study on the effect of stalk and root lodging resistance of plants grown with and without film. The chosen methods and study methodology are adequate to the research scopes analyzed. The results obtained are related to the literature of other authors.

The results obtained contribute to the theoretical basis and technical support for successful planting of maize without foil.

I accept the article in its present form.

Author Response

Dear reviewer,

Thanks for the recognition on our manuscript “Maize Lodging Resistance with Plastic Film Removal, Increased Planting Density, and Cultivars with Different Maturity Periods” (No: plants-1956231).